# Relationship between Cognitive Strategies of Emotion Regulation and Dimensions of Obsessive–Compulsive Symptomatology in Adolescents

**DOI:** 10.3390/children10050803

**Published:** 2023-04-29

**Authors:** Jesús Ferrández-Mas, Beatriz Moreno-Amador, Juan C. Marzo, Raquel Falcó, Jonatan Molina-Torres, Matti Cervin, José A. Piqueras

**Affiliations:** 1Department of Health Psychology, Miguel Hernández University of Elche, 03202 Alicante, Spain; 2Department of Clinical Sciences Lund, Lund University, Box 117, 221 00 Lund, Sweden; matti.cervin@med.lu.se; 3Child and Adolescent Psychiatry, Skåne, Sofiavägen 2E, 221 85 Lund, Sweden

**Keywords:** OCD, OCD symptomatology dimensions, emotion dysregulation, cognitive-emotion regulation, adolescents

## Abstract

Cognitive emotion regulation refers to the management of one’s emotions through cognitive strategies. Studies have found that individuals with obsessive–compulsive symptoms utilize emotion regulation strategies differently compared to those without these symptoms. This study aims to investigate the relationship between cognitive strategies for emotion regulation and specific dimensions of obsessive–compulsive symptoms in adolescents. A cross-sectional descriptive study was conducted with 307 adolescents between 12 and 18 years old. Associations between sociodemographic variables, obsessive–compulsive symptoms, and emotion regulation strategies were examined using regression and network analyses. Regression results indicated that emotion regulation strategies and gender accounted for 28.2% of the variation in overall obsessive–compulsive symptoms (*p* < 0.001) and that emotion regulation explained most variance in the symptom dimension of obsessing. Network analysis showed that self-blame and catastrophizing were uniquely linked to overall obsessive–compulsive symptoms, while several strategies were uniquely linked to specific symptom dimensions. The adaptive strategy that demonstrated the strongest association with obsessive–compulsive symptoms was refocus on planning, while maladaptive strategies included catastrophizing, self-blame, and rumination. In conclusion, the results support the relationship between cognitive strategies for emotion regulation and dimensions of obsessive–compulsive symptoms in adolescents, though these relations appear complex and require further investigation. Addressing emotion regulation in the prevention of obsessive–compulsive symptoms may be warranted, but prospective studies are needed.

## 1. Introduction

Obsessive Compulsive Disorder (OCD) is a mental health condition that is characterized by the presence of obsessions and compulsions. Obsessions are recurrent thoughts, images, or impulses that the patient experiences as unwanted and intrusive, while compulsions are repetitive behaviors or mental events that the person feels the urge to do in response to an obsession [1]. In the Diagnostic and Statistical Manual of Mental Disorders (DSM-5) of the APA (2013), OCD is no longer considered an anxiety disorder and has been included in a separate chapter called “Obsessive–compulsive and related disorders”, which also includes body dysmorphic disorder, hoarding disorder, trichotillomania (hair pulling disorder), and excoriation (skin picking) disorder. 

The content of obsessions and compulsions differs among people with OCD. Some symptom dimensions tend to be common, such as contamination and cleanliness, symmetry and ordering, and forbidden or harmful thoughts [1]. A meta-analysis suggested four symptom dimensions: forbidden thoughts, hoarding, symmetry, and contamination/cleaning [2]. In contrast, one of the most commonly recognized classifications of OCD dimensions is that which differentiates between doubt/check, obsession, neutralization, tidiness, washing, and hoarding [3,4,5]. However, it seems that there is still a lack of agreement on the dimensionality of OCD symptoms, and there may be more dimensions than previously thought. In this regard, it is noteworthy that a study by Cervin et al. [6] found that OCD symptoms can be grouped into the following eight dimensions: disturbing thoughts, incompleteness, contamination, accumulation, transformation, body focus, superstition, and loss/separation. These dimensions were valid throughout an individual’s lifespan, and incompleteness and disturbing thoughts were most prominent, being uniquely associated with many other dimensions. Moreover, the presence of these dimensions has been confirmed in both clinical and non-clinical adolescent populations [7]. 

Cognitive Behavioral Therapy (CBT), particularly the Exposure and Response Prevention technique (ERP), has been identified as the most effective psychological intervention for treating OCD [8,9,10]. These treatments are effective in both children and adolescents [11]. Nonetheless, more recent studies suggest that augmenting the core ERP approach with self-compassion and emotional regulation components may yield additional benefits [12]. It is also important to note here that the prevention of OCD has received limited focus, and several studies highlight the need to identify children and adolescents at risk of developing this disorder by means of early detection as well as selective and indicated preventive interventions [13,14].

Therefore, the evidence regarding the use of psychological interventions for childhood OCD is at a point where we are attempting to improve treatment by including new strategies or processes involved in the maintenance of the disorder. In this sense, an improvement in emotional regulation skills would lead to an improvement in the treatment of OCD; therefore, psychological interventions for OCD should consider the possibility of employing techniques aimed at emotion regulation, such as Acceptance and Commitment Therapy (ACT) [15]. This is consistent with the findings of several studies that indicate that people with OCD have difficulties with emotion regulation [16,17,18,19,20], which has been confirmed in both clinical and non-clinical populations and after controlling for the effects of depression and anxiety [15]. Although most studies on emotion regulation and OCD have been carried out in the adult population, the relationship between emotion dysregulation and OCD has also been found in adolescents with OCD at an even greater magnitude than that observed between emotion dysregulation and anxiety disorders [21].

Emotion regulation is considered a process of modulation and adaptive response to one’s emotions [22]. Emotion regulation includes a broad set of antecedent- and response-focused strategies, of which cognitive reappraisal (antecedent-focused) and expressive or emotional suppression (response-focused) have been most studied [23]. There is some agreement that cognitive reappraisal is a more effective or adaptive strategy than expressive emotional suppression, which can be self-defeating [17]. Other authors define emotion regulation as a multidimensional construct that includes emotional awareness, emotional clarity, emotional acceptance, access to emotional regulation strategies, impulse control, and goal-directed behaviors [24]. There seems to be no consensus on which constructs to include when talking about emotion regulation [25].

In this regard, one type of emotion regulation strategy is related to cognitive emotion regulation, which is the way we regulate emotions through thoughts or cognitions, i.e., how we think about emotions in daily life [26,27,28]. This concept is closely related to the construct of cognitive coping, although there is an important difference between them, as emotion and problem-focused coping strategies include a combination of both cognitive and behavioral mechanisms, whereas cognitive emotion regulation assumes that thought and behavior are distinct processes and therefore considers cognitive strategies purely conceptually, independently of behavioral strategies [27]. 

Cognitive coping or the cognitive components of emotion regulation have not been studied in depth, and are isolated so far from the other dimensions of coping. As a result, although there has been substantial interest in cognitive processes as regulators, little is known about how cognitions regulate emotions and how this may affect the course of emotional development [29]. Some studies suggest there are nine cognitive strategies of emotion regulation, which can be divided into adaptive and maladaptive forms of regulation [26]. The adaptive strategies include acceptance, positive refocusing, putting into perspective, positive reappraisal, and refocus on planning, while the maladaptive strategies include self-blame, rumination, catastrophizing, and other-blame.

Patients with OCD have been shown to experience a higher frequency of intrusive thoughts after using emotional suppression compared to acceptance [30]. Likewise, maladaptive emotion regulation, such as that which occurs with expressive emotional suppression, is related to greater cognitive rigidity and obsessive–compulsive symptoms [18]. In this regard, Berman et al. [31] observed that the expressive emotional suppression strategy showed a positive association with obsessive beliefs and the severity of obsessive–compulsive symptoms in children and adolescents. It was also observed that expressive emotional suppression had a positive association with all dimensions of OCD except hoarding. Other authors have observed that non-acceptance of emotions and difficulties in goal-directed behaviors showed unique associations with OCD [15]. People with OCD generally have greater difficulty understanding and tolerating their emotional states, according to a review study [25]. Furthermore, this study showed that improved tolerance of difficult internal experiences was associated with a decrease in OCD symptoms. In the same vein, data suggest that people with OCD may use compulsions as an attempt to regulate the distress caused by obsessions, thus fulfilling an emotion regulation function [32]. However, these compulsions would be considered a maladaptive emotion regulation mechanism.

Concerning adaptive emotion regulation strategies and OCD, a systematic review noted that cognitive reappraisal and acceptance strategies could reduce distress and intrusive thoughts in patients with OCD [17], and reappraisal was related to increased positive affect and reduced OCD symptoms [18]. In contrast, distress in OCD patients is reduced when acceptance strategies are used [30]. This study also found that distress was also reduced when distraction was used, although findings on the long-term efficacy of this strategy were inconsistent. Other authors have observed that adaptive emotion regulation has a negative association with obsessive beliefs and some dimensions of OCD, especially washing symptoms, in child and adolescent populations. Thus, it might be appropriate to teach adaptive emotional regulation skills to youth with washing symptoms before and during exposure intervention [31]. 

Although in recent years there has been a considerable increase in the number of published studies on the relationship between emotion regulation and OCD, there have been few studies carried out with adolescents. As adolescence is considered a risk period for the onset of obsessive–compulsive symptoms and the development of OCD [33] and is a critical phase in the development of emotion regulation mechanisms [34], the present study aims to examine the relationship between cognitive strategies for emotion regulation and specific dimensions of obsessive–compulsive symptoms. We examine the relationship between cognitive strategies of emotion regulation and OCD symptoms in a sample of adolescents. According to the literature, we hypothesize that adaptive cognitive emotion regulation strategies will be negatively related to obsessive–compulsive symptoms, whereas maladaptive strategies will be positively related to obsessive–compulsive symptoms.

## 2. Materials and Methods

### 2.1. Participants 

A total of 307 adolescents from high school in the province of Alicante participated in this study, of whom 56.4% (*n* = 173) identified themselves as female, 43% (*n* = 132) as male, 0.3% (*n* = 1) as intersex, and 0.3 (*n* = 1) did not wish to report their gender. The age of individuals in the sample ranged between 12 and 18 years, and the mean age was 14.78 years (*SD* = 1.48).

This is a non-probability convenience sample. The inclusion criteria were to be between 12 and 18 years of age and to provide written consent from both the student and the parents/legal guardians. The exclusion criteria were impaired cognitive functioning and/or poor comprehension of the Spanish language that prevented correct completion of the self-report questionnaires. To check exclusion criteria related to the presence of reading comprehension or cognitive functioning problems, the official reports from the educational centres were used. These reports contain information about special educational needs of the pupils. Nevertheless, all participants who submitted consent were included in the administration of the survey in the classroom, adapting it to their level of proficiency, receiving support from the research team, or proposing alternative activities, but their data were not used. Participation in the study was anonymous and voluntary and there were no rewards for collaboration. 

### 2.2. Instruments and Variables 

#### 2.2.1. Socio-Demographic Data 

Socio-demographic data were collected through a short ad hoc questionnaire designed to collect data on age, sex, gender identity, sexual orientation, level of education, academic year, and country of birth.

#### 2.2.2. Obsessive–Compulsive Symptoms

Obsessive–compulsive symptoms were assessed using the Obsessive–Compulsive Inventory—Child Version (OCI-CV) [35]. It is a self-report questionnaire consisting of 21 items, with a 3-point Likert-type scale (0 = “never”, 1 = “sometimes”, 2 = “always”). The total score ranges from 0 to 42. It is designed to evaluate OCD symptoms in children and adolescents and measures the following factors: doubting/checking, obsessing, hoarding, washing, ordering, and neutralizing. The total score of the scale presents high internal consistency (α = 0.81), as well as the subscales (α = 0.81–0.88), and moderate intercorrelations between the different subscales [35]. The Spanish version of the OCI-CV has shown evidence of test–retest reliability and convergent and discriminant validity [5,36]. 

#### 2.2.3. Emotion Regulation Strategies 

The Spanish adaptation of the Cognitive Emotion Regulation Questionnaire (CERQ) [26] was used to assess emotion regulation strategies [37]. It is a questionnaire composed of 36 items that assess nine cognitive strategies of emotion regulation: self-blame, acceptance, rumination, positive focus, a reorientation towards planning, positive reappraisal, putting into perspective, catastrophizing, and blaming others. Each of these subscales is composed of four items, with a 5-point Likert-type response format ranging from 1 (“almost never”) to 5 (“almost always”). The scores for each subscale are obtained by summing the scores for the items of each subscale and can range from a minimum of 4 to a maximum of 20. The questionnaire has reported strong psychometric properties in the adolescent population, with an internal consistency of the subscales ranging from α = 0.68 to α = 0.83 [26]. It has also shown evidence of reliability and validity both in its original version [26] and in Spanish adolescents [37,38]. Finally, it should be noted that the nine dimensions that make up the CERQ could be grouped into two more general categories, which some authors referred to as ‘adaptive strategies’ and ‘maladaptive strategies’ [26,39]. The adaptive strategies would include the dimensions of acceptance, positive refocusing, putting into perspective, positive reappraisal, and refocusing on planning. In contrast, maladaptive strategies would include self-blame, rumination, catastrophizing, and other-blame.

### 2.3. Procedure 

The study was approved by the research and ethics committees and Project Evaluation Body of the University Miguel Hernandez of Elche, with reference number “DPS.JPR.02.17”. Once the project was approved, the educational institution was contacted to inform them of the objective of the study and the procedure that was going to be used for evaluation. Likewise, they were informed of the data protection and confidentiality regulations, and informed written consent was requested from the participants and their parents or legal guardians in the case of minors. Data collection was carried out by the research staff in a classroom at the school during school hours. The questionnaires were applied in a group format and were completed online using the “Lime Survey” software. The assessment duration was about 10 min.

#### 2.3.1. Type of Design 

This is a descriptive, cross-sectional, correlational study.

#### 2.3.2. Data Analysis 

The data were coded and analyzed using SPSS Statistics (version 25.0), with a confidence level of 95%. Descriptive analyses were carried out, including frequencies with percentages and mean with standard deviations to determine the characteristics of the sample. Next, reliability analyses of the scales assessing the variables “OCD symptoms” and “emotion regulation strategies” were carried out, calculating Cronbach’s alpha coefficient.

In addition, the Student’s *t*-test for independent samples was used to analyze whether there were statistically significant differences according to gender concerning the manifestation of OCD symptomatology. Cohen’s criteria was used for the interpretation of effect size, with *d* = 0.20–0.49 being a small effect size, *d* = 0.50–0.79 a moderate effect size, and d greater than or equal to 0.80 a large effect size [40].

Analyses were performed to verify whether the study sample met the assumptions of multiple linear regression. It was confirmed that the assumptions of independence, linearity, and homoscedasticity were fulfilled. However, it was observed that the assumption of normality was not met. Therefore, non-parametric correlations with Spearman’s rho were calculated to analyze the association between the different variables as it was observed that the sample did not follow a normal distribution. Interpretation of the magnitude of the correlations was carried out following the criteria proposed by Cohen [40], which suggest that a value between 0.10 and 0.29 indicates a small correlation, between 0.30 and 0.49 is a moderate correlation, and above 0.50 is a large one. Also, bivariate correlations were checked for values below 0.70 to rule out multicollinearity. 

Finally, once the main assumptions of the multiple regression model were verified, multiple regression analyses were performed to explore the relationship between cognitive strategies for emotional regulation and OCD symptomatology. The sociodemographic variable sex was included in the predictive models to control for its effect on OCD symptomatology. The coefficient of determination (R2), standardized beta coefficients (β), and semi-partial correlation coefficient (sr2) were calculated to explore the specific contribution of each variable in terms of the percentage of variance explained.

Lastly, to parse out the unique associations among the full set of variables, network analysis was used. In network analysis, the unique association between each variable pair is estimated while accounting for all other linear associations in the full set of variables. Analyses were conducted in R Studio using the R library BGGM. Unique associations were estimated as partial correlation coefficients ranging from −1 to +1 using a semi-parametric copula model based on ranked likelihood, which can handle non-normal variables as well as a mix of non-normal and normal variables. To control for false positive rate, we used 95% credible intervals for each unique association. A 95% credible interval indicates where a population parameter will fall 95% of the time. Credible intervals that did not include zero were considered statistically significant, and these associations were plotted as a graphical network using the Fruchterman–Reingold algorithm implemented in the R-package *qgraph,* which places nodes with many unique associations to other nodes centrally while placing strongly associated node pairs closely. Age and gender were included in the network to account for their potential effects.

## 3. Results

Table 1 shows the descriptive statistics of the variables studied, as well as the internal consistency indices found through the reliability analysis for each of the scales and subscales.

Age was not statistically significantly associated with OCD symptomatology. However, statistically significant differences were observed between gender and the criterion variable (*rho* = 0.27; *p* < 0.001). Therefore, analyses were performed using a Student’s t-test for independent samples (equal variances were not assumed) to explore whether being male or female influenced the manifestation of OCD symptomatology. Females scored significantly higher than males on the OCI-CV total, *t*(289.7) = −4.867, *p* < 0.001, with a Cohen’s d magnitude of differences = −0.57, meaning that there is a medium effect size. Therefore, it was decided to include gender as a covariate in the multiple regression analyses.

### 3.1. Associations

Table 2 presents the correlations observed between the predictor variables, which were the emotion regulation strategies and the OCD symptomatology criterion variable. Correlations with the different subscales of the OCI-CV are also included. 

As can be seen, statistically significant differences were found between all predictor variables and OCD symptomatology, except the acceptance and putting into perspective subscales. The variables positive refocusing (rho = −0.22; *p* < 0.01), refocus on planning (rho = −0.20; *p* < 0.01), and positive reappraisal (rho = −0.268; *p* < 0.01) showed a low inverse correlation with the criterion variable. It can also be observed that the variables self-blame (rho = 0.29; *p* < 0.01) and rumination (rho = 0.24; *p* < 0.01) showed a low positive correlation with the criterion variable, while blaming others (rho = 0.18; *p* < 0.01) showed a very low positive correlation. Catastrophizing (rho = 0.42; *p* < 0.01) showed the highest correlation with OCD symptomatology, with a moderate positive relationship. Regarding the subscales of the OCI-CV, the predictor variable acceptance did not show statistically significant differences with any of the subscales. In contrast, the variables positive refocusing, refocus on planning, and positive reappraisal showed low and very low negative correlations with the subscales obsessing, hoarding, doubting, and neutralizing. The putting into perspective variable only showed statistically significant differences with the neutralizing subscale, with a very low negative correlation. In contrast, concerning maladaptive strategies, the variable self-blame showed positive correlations of low and very low size with all subscales, except with hoarding, where no statistically significant differences were observed. Low and very low positive correlations of size were also observed between rumination and the subscales of the OCI-CV, except for hoarding and neutralizing, where no statistically significant differences were observed. The variable catastrophizing presented statistically significant differences with all subscales of the OCI-CV, with a positive correlation of low size between them, except in the case of hoarding, where the size was very low, and obsessing, which was moderate. Finally, the variable other-blame only showed statistically significant positive correlations of very low size with the neutralizing subscale and low size with order.

### 3.2. Predictive Models 

Linear regression models were used to predict global OCD symptomatology and each symptom dimension of OCI-CV (Table 3). In the model created to predict global OCD symptomatology, it was observed that about 28.2% (*p* < 0.001) of the variance in the criterion variable was explained by CERQ emotion regulation strategies and gender. It should be noted that statistically significant contributions were found for the variables gender (sr2 = 2.9%), refocus on planning (sr2 = 1.2%), self-blame (sr2 = 3%), and catastrophizing (sr2 = 3.9%). The relationship between global OCD symptomatology and the variables gender, self-blame, and catastrophizing was direct, while the relationship with refocus on planning was inverse. This suggests that lower use of the refocus on planning strategy would be associated with greater OCD symptomatology. However, it should be noted that all models created for the prediction of the OCI-CV subscales based on emotion regulation strategies and gender presented statistically significant contributions in explaining the variance. In the case of the gender variable, a significant contribution was found in all dimensions of the OCI-CV except for washing. As for the emotion regulation strategies, in the obsessing subscale, significant inverse contributions were found for the variables positive refocusing and refocus on planning, with direct contributions for self-blame, rumination, and catastrophizing. In the washing subscale, there were significant contributions of the variables acceptance (inversely), and of self-blame, rumination, and catastrophizing (directly). Hoarding showed a significant contribution from refocus on planning with an inverse relationship. Doubting had significant contributions from the variables of self-blame and catastrophizing (directly). Neutralizing showed significant contributions of the variables refocus on planning (inversely) and self-blame (directly). Finally, ordering had significant direct contributions from the variables catastrophizing and other-blame.

### 3.3. Network Analyses

Networks for symptom dimensions and the total OCI-CV score and emotion regulation variables are presented in Figure 1. For symptom dimensions, rumination was significantly linked to obsessing (partial *r* = 0.13 [0.00, 0.26]) and washing (partial *r* = 0.16 [0.04, 0.29]), planning was linked to ordering (partial *r* = 0.13 [0.00, 0.25]) and obsessing (partial *r* = −0.13 [−0.00, 0.25]), other-blame was linked to ordering (partial *r* = 0.18 [0.06, 0.30]), catastrophizing was linked to obsessing (partial *r* = 0.21 [0.09, 0.33]), and perspective was linked to neutralization (partial *r* = −0.15 [−0.01, −0.29]). When the total score of OCI-CV was analyzed, catastrophizing (partial *r* = 0.25 [0.13, 0.36]) and self-blame (partial *r* = 0.22 [0.10, 0.33]) were the two variables with a significant link to obsessive–compulsive symptoms. Regarding the associations between female gender and obsessing and hoarding, the partial correlations were 0.25 and 0.19, respectively.

## 4. Discussion

This study aimed to analyze the relationship between cognitive emotion regulation strategies and OCD symptoms in a sample of Spanish adolescents. According to previous literature, the hypothesis was that adaptive cognitive emotion regulation strategies would be negatively related to the presence of OCD symptomatology in adolescents, whereas maladaptive strategies would show a positive relationship with OCD symptomatology.

Although the aim of this study was not to analyze gender differences, these differences were examined in the preliminary analyses to consider their effect on the hypothesis. Thus, it was observed that women had higher scores than men in the OCI-CV total scores and the different subscales, except for washing, where no statistically significant differences were found. These results were replicated in network analyses, where female gender was significantly associated with higher scores on OCI-CV, particularly the obsessing and hoarding dimensions, but with a small to moderate effect size. These findings are consistent with several studies conducted in adolescent populations [41,42,43,44]. However, other studies did not find the gender differences in adolescents [15,36,45,46]. Therefore, it could be argued that this is a topic on which there is some controversy, and more research is still needed to reach a consensus.

The results showed that the adaptive emotion regulation strategies that presented a statistically significant negative correlation with global OCD symptomatology were positive refocusing, refocus on planning, and positive reappraisal, with a small effect size in all cases. In the model created to predict global OCD symptomatology, the refocus planning strategy was the only adaptive strategy that showed a unique contribution to the prediction of variance, and in the network models, no adaptive strategy was uniquely linked to the total score of the OCI-CV. In the linear regression models, in relation to each of the OCI-CV dimensions, refocus on planning contributed to the explanation of the dimensions of obsessing, hoarding, and neutralizing. Although we know of no previous study that would allow us to compare the relationship between the use of this strategy and OCD symptomatology, there are similarities with the results of other studies that have found a negative relationship between refocus on planning and symptoms of anxiety and depression in adolescents [47]. 

Nevertheless, we also found a contribution of the positive focusing strategy in explaining the obsessing dimension. According to Garnefski et al. [26], this strategy involves thinking about pleasant things instead of the event itself, and this might be a useful response in the short term but may prevent more adaptive coping in the long term. In this way, it could be understood as a form of distraction from the stressful event, consistent with research that found that people with OCD reduced the distress caused by their obsessive thoughts by using the distraction strategy, although the long-term effectiveness of this strategy could not be demonstrated [30]. It should be noted that other authors also observed no relationship between the positive focusing strategy and anxiety symptomatology in an adolescent population [47]. 

A unique negative contribution of the acceptance strategy was also found in the washing dimension. These results are in the same line as other studies that point to the importance of adaptive emotion regulation in contamination and washing symptoms [31]. However, it is relevant that no contribution was found with the rest of the subscales of the OCI-CV, considering the recent literature pointing out that the use of acceptance is related to a reduction in distress and the frequency of obsessive thoughts [17]. In this regard, it should be noted that although acceptance is considered in the CERQ as an adaptive strategy [26], in some studies that have used this measure, positive correlations with depression have been observed [27,48]. Therefore, according to some authors [38], it might be interesting to review and further clarify this strategy in the CERQ or to differentiate between active and passive acceptance, as the latter could be interpreted negatively as a form of resignation. 

The positive reappraisal strategy did not contribute to explaining any of the OCI-CV dimensions or overall obsessive–compulsive symptoms, either in regression or in network models, which is consistent with previous studies in adolescents [31] and the adult population [49]. However, these results contrast with the review conducted by Ferreira et al. [17], where a relationship between cognitive reappraisal and a reduction in OCD symptoms was observed, although it should be noted that the sample consisted of an adult population and included patients with clinical OCD. 

The putting into perspective strategy did not show any contribution to the models created to predict OCD symptomatology using regression but was negatively linked to neutralizing in the network model. The latter finding should be interpreted with caution due to the high probability of spurious negative associations in network models with many variables. Overall, our results are consistent with a recent study that found no differences between people with OCD and healthy individuals regarding perspective taking, although in that study they used other measures to assess these variables [50]. Other research in adolescent populations also found no association between putting in perspective and other disorders such as depression and anxiety [47,51].

Regarding maladaptive emotion regulation strategies, it was observed that all of them presented a statistically significant positive correlation with small to moderate effect sizes, with global OCD symptomatology in separate regression models; catastrophizing and self-blame strategies showed unique contributions in the regression and network models where the total OCI-CV score was used. Likewise, the catastrophizing strategy contributed to the models for the subscales of obsessing, washing, doubting, neutralizing, and ordering (using regression), and obsessing (using network modeling). These results are consistent with the research of Paul et al. [19], which showed that OCD patients had higher scores on the catastrophizing subscale of the CERQ. Moreover, the role of catastrophizing in other disorders such as anxiety and depression has been reported on several occasions [47,52]. Conversely, self-blame also showed a unique contribution in models for the prediction of obsessing, washing, doubting, and neutralizing dimensions. Although no research has specifically studied the relationship between this strategy and OCD symptoms, there is evidence that self-blame is related to the manifestation of anxiety symptomatology in adolescents [47], and in both anxiety and depression in childhood [53] and the general population [52]. 

Regarding the rumination strategy, a unique contribution was found in the models for predicting the obsessing and washing dimensions, and rumination was the only strategy that was uniquely linked to two dimensions (obsessing and washing) in the network model with the OCI-CV dimensions. These results are consistent with the limited research available on this topic that establishes a relationship between rumination and obsessive–compulsive symptoms [54,55], although in the present study, no significant contribution of rumination to the OCI-CV total score was found. Even though different measurement instruments were used, the relationship between rumination and the washing dimension is relevant, although these authors observed relationships with a larger number of dimensions [55]. It should be noted that other researchers have also analyzed this relationship, finding only associations between rumination and the dimensions of responsibility for harm [56] and neutralizing [57]. Considering the limited research on this topic and the lack of concordance in the results, further research is considered necessary to understand the relationship between rumination and OCD symptomatology.

The strategy of other-blame showed a unique contribution to the model in predicting the ordering dimension, using both regression and network models. As with the other strategies assessed, no research has been found that specifically addresses the relationship between other-blame and OCD symptomatology. However, it should be noted that a relationship has been found between this strategy and symptoms of anxiety and depression in adolescents [46] and the general population [52].

The results obtained show that cognitive strategies of emotion regulation are associated with the presence of OCD symptoms in adolescents. These findings are consistent with previous research indicating that there is a relationship between emotion regulation difficulties and OCD symptomatology in adolescents [21]. As adolescence is considered a risk period for the development of obsessive–compulsive disorder [33] and is a key stage in the acquisition of emotion regulation strategies [34], knowing the strategies that are most clearly associated with OCD can be very useful for the development of prevention and treatment programs in adolescents. Prevention strategies aimed at increasing emotional stability would have the greatest benefits in reducing obsessive–compulsive symptomatology if they were applied before the age of 16 [43], so it would be interesting to implement these programs during the early years of adolescence, although prospective research is needed to identify the casual relations between symptoms and emotion regulation strategies.

The study has some limitations that need to be considered. Firstly, it is a cross-sectional study, so the predictive validity of the models is unknown. Indeed, alternative explanations for our findings include the possibility that cognitive strategies of emotion regulation are a consequence of OCD symptoms, rather than a predisposing factor. Further studies that have found that cognitive strategies of emotion regulation play a role in symptom coping have used clinical samples with OCD, not young people with OCD symptoms, as in our case [44]. In this sense, it would be interesting for future research to study these variables through longitudinal studies, which are currently missing. Another limitation is the use of self-reports, as this type of measure does not allow clinical diagnoses to be obtained and can also lead to the appearance of social desirability bias and false answers. Although we tried to minimize these limitations through anonymous and voluntary participation, it may be beneficial for future research to include clinical interviews as well. It should also be noted that 53.7% of the sample exceeded the clinical cut-off point suggested by other authors in a US population [46]. Given the cultural, linguistic, and demographic differences between the two samples, these cut-off points may not be directly relevant to our sample. Furthermore, this could indicate problems with the cross-cultural validity of the questionnaire and indicates the need for additional studies that establish clinical cut-off points in the Spanish population, specifically in adolescents. Nonetheless, as the sample was non-probabilistic and the number of participants was small, it is not possible to generalize the results. Therefore, it would be interesting for future studies to use a random sampling method in a clinical population, which would allow for generalization of the results and more solid conclusions about the relationship between emotion regulation and OCD. Finally, it should be noted that due to the limited number of studies that have used the CERQ to assess emotion regulation strategies concerning OCD symptomatology, comparison of our results with other research has been limited.

## 5. Conclusions

The main contribution of this study has been to examine the relationship between each of the cognitive strategies of emotion regulation and the different dimensions of OCD in adolescents. We observed that the adaptive strategy of refocus on planning and other less adaptive strategies such as the predisposition to catastrophizing, self-blame, and rumination could have relevant roles in the development and/or maintenance of OCD and that inclusion of these strategies in prevention and treatment programs for OCD in adolescents could be beneficial.

## Figures and Tables

**Figure 1 children-10-00803-f001:**
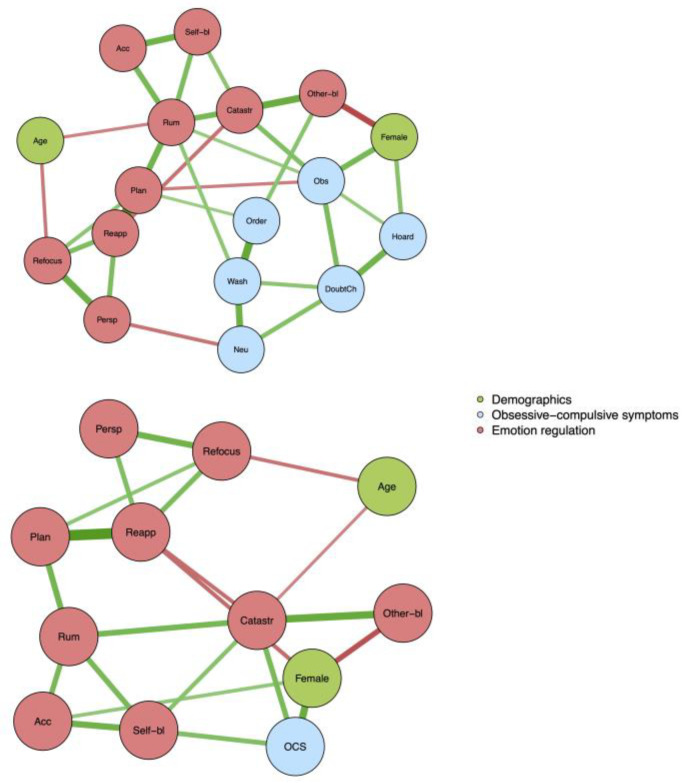
Networks for symptom dimensions and the total OCI-CV score and emotion regulation variables.

**Table 1 children-10-00803-t001:** Descriptive statistics of the sample.

	Total Sample (*n* = 307)			Men (*n* = 132)		Women (*n* = 173)	
	α	Mean	(SD)	Mean	(SD)	Mean	(SD)
OCI-CV Total	0.89	12.01	(7.10)	9.85	(6.64)	13.68	(7.04)
OCI-CV Obsessing	0.78	2.49	(1.96)	1.78	(1.71)	3.02	(1.98)
OCI-CV Washing	0.66	1.97	(1.53)	1.83	(1.51)	2.09	(1.54)
OCI-CV Hoarding	0.68	1.63	(1.40)	1.22	(1.21)	1.95	(1.46)
OCI-CV Doubting	0.76	2.91	(2.13)	2.45	(2.01)	3.27	(2.15)
OCI-CV Neutralizing	0.57	0.68	(1.01)	0.55	(0.90)	0.79	(1.07)
OCI-CV Ordering	0.65	2.33	(1.57)	2.02	(1.48)	2.58	(1.60)
CERQ Acceptance	0.65	13.50	(3.37)	13.05	(3.33)	13.86	(3.38)
CERQ Positive refocusing	0.85	12.71	(4.49)	13.42	(4.02)	12.17	(4.79)
CERQ Refocus on planning	0.78	14.76	(3.46)	15.19	(3.20)	14.46	(3.64)
CERQ Positive reappraisal	0.80	14.43	(3.84)	15.30	(3.24)	13.79	(4.14)
CERQ Putting into perspective	0.71	13.63	(3.65)	13.91	(3.73)	13.43	(3.60)
CERQ Self-blame	0.68	10.56	(3.30)	10.30	(3.27)	10.75	(3.34)
CERQ Rumination	0.74	12.60	(3.63)	12.26	(3.79)	12.86	(3.52)
CERQ Catastrophizing	0.71	9.14	(3.61)	9.05	(3.63)	9.19	(3.62)
CERQ Other-blame	0.79	8.63	(3.46)	9.28	(3.40)	8.09	(3.43)

Note. α = Cronbach’s α; *SD* = standard deviation; OCI-CV = Obsessive–Compulsive Inventory-Children’s Version; CERQ = Cognitive Emotion Regulation Questionnaire.

**Table 2 children-10-00803-t002:** Spearman correlation coefficients between the total score and subscales of the OCI-CV and the CERQ dimensions.

Variable	1	2	3	4	5	6	7	8	9	10	11	12	13	14	15	16
1. OCI-CV Total	1															
2. Obsessing	0.76 **	1														
3. Washing	0.69 **	0.36 **	1													
4. Hoarding	0.65 **	0.44 **	0.26 **	1												
5. Doubting	0.83 **	0.57 **	0.46 **	0.52 **	1											
6. Neutralizing	0.63 **	0.37 **	0.47 **	0.36 **	0.45 **	1										
7. Ordering	0.73 **	0.41 **	0.54 **	0.35 **	0.47 **	0.42 **	1									
8. Acceptance	0.03	0.11	−0.05	0.01	0.07	−0.07	−0.01	1								
9. Refocusing	−0.22 **	−0.25 **	−0.08	−0.17 **	−0.21 **	−0.12 *	−0.11	0.23 **	1							
10. Planning	−0.20 **	−0.22 **	−0.03	−0.24 **	−0.19 **	−0.20 **	−0.04	0.28 **	0.51 **	1						
11. Reappraisal	−0.27 **	−0.30 **	−0.05	−0.22 **	−0.27 **	−0.20 **	−0.10	0.25 **	0.56 **	0.69 **	1					
12. Perspective	−0.08	−0.09	−0.02	−0.10	−0.05	−0.15 **	−0.03	0.30 **	0.48 **	0.43 **	0.48 **	1				
13. Self-blame	0.29 **	0.33 **	0.19 **	0.11	0.27 **	0.12 *	0.17 **	0.46 **	0.09	0.21 **	0.11	0.22 **	1			
14. Rumination	0.24 **	0.29 **	0.18 **	0.05	0.18 **	0.07	0.17 **	0.43 **	0.09	0.31 **	0.12 *	0.22 **	0.54 **	1		
15.Catastro-phizing	0.42 **	0.42 **	0.23 **	0.17 **	0.33 **	0.28 **	0.32 **	0.12 *	−0.04	−0.01	−0.15 *	0.07	0.38 **	0.43 **	1	
16. Other-blame	0.18 **	0.11	0.10	0.09	0.11	0.18 **	0.25 **	−0.04	0.01	0.01	0.01	0.10	0.08	0.21 **	0.43 **	1

Note. * *p* < 0.05; ** *p* < 0.01.

**Table 3 children-10-00803-t003:** Multiple linear regression models for total OCD symptomatology and subscales of the OCI-CV.

	Obsessive–Compulsive Inventory: Children’s Version													
	Total		Obsessing		Washing		Hoarding		Doubting		Neutralizing		Ordering	
Variable	*β*	*t*	*β*	*t*	*β*	*t*	*β*	*t*	*β*	*t*	*β*	*t*	*β*	*t*
Gender	0.18	3.58 **	0.24	4.97 **	0.08	1.36	0.23	4.10 **	0.14	2.55 *	0.13	2.24 *	0.19	3.45 *
Acceptance	−0.06	−0.94	−0.01	−0.01	−0.17	−2.52 *	0.03	0.50	−0.04	−0.60	−0.07	−1.08	−0.07	−1.10
Refocusing	−0.09	−1.44	−0.12	−2.08 *	−0.03	−0.40	−0.04	−0.55	−0.07	−1.04	0.05	0.72	−0.07	−0.99
Planning	−0.16	−2.25 *	−0.19	−2.85 *	−0.09	−1.17	−0.18	−2.26 *	−0.14	−1.84	−0.17	−2.18 *	0.02	0.27
Reappraisal	−0.03	−0.41	−0.04	−0.50	0.10	1.16	−0.04	−0.46	−0.09	−1.09	0.03	0.32	−0.01	−0.10
Perspective	−0.03	−0.45	0.01	0.06	−0.05	−0.66	0.01	0.03	−0.01	−0.18	−0.13	−1.91	−0.01	−0.18
Self-blame	0.23	3.62 **	0.19	3.21 *	0.20	2.78 *	0.10	1.41	0.25	3.83 **	0.16	2.30 *	0.09	1.30
Rumination	0.07	1.04	0.13	2.12 *	0.15	2.00 *	−0.01	−0.21	0.01	0.14	−0.06	−0.81	0.01	0.10
Catastrophizing	0.26	4.09 **	0.27	4.47 **	0.15	2.17 *	0.08	1.19	0.20	3.02 *	0.23	3.29 *	0.20	2.90 *
Other-blame	0.07	1.22	0.01	0.12	0.01	0.11	0.10	1.59	0.03	0.46	0.13	2.15	0.18	2.85 *
R^2^	0.282		0.375		0.095		0.121		0.207		0.142		0.135	
F	13.04 **		17.66 **		4.18 **		5.17 **		8.94 **		6.05 **		5.73 **	

Note. R^2^ = Coefficient of determination; F = F-index corresponding to R^2^; *β* = standardized beta; *t* = Student’s t-test; * *p* < 0.05; ** *p* < 0.001.

## Data Availability

Data are available at the request of the corresponding author.

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
