# Peer review of "Relationship between Cognitive Strategies of Emotion Regulation and Dimensions of Obsessive–Compulsive Symptomatology in Adolescents"

_children, 2023, doi:10.3390/children10050803_

Round 1

Reviewer 1 Report

This is an interesting study, where the authors try to relate emotion regulation strategies to OCD symptomatology. A general problem I see with the study is that the items listed as emotion regulation strategies (especially the maladaptive ones) might be related to OCD itself, and not entirely independent of it. It may be that participants may be restricted in using a certain emotion regulation strategy by OCD symptom severity. Not that they have preferentially this or that emotion regulation style but they might be restricted in using this or that emotion regulation strategy by the diagnosis itself. In other words, the reasoning might be circular. 

Minor points

1.     Line 9 abstract: The authors write in the abstract “Emotion regulation refers to the management of one's emotions by cognitive strategies.” But not all emotion regulation strategies are cognitive. There is also implicit emotion regulation, suppression, avoidance are other strategies may not always be cognitive strategies. They qualify this later in the text, but it should be reformulated here to reflect the complexity.  

2.     Line 54: The authors write on line 54 that “The average age of the onset of OCD is 19 years old, and around 25% of cases frequently begin at the age of 14 years” Is this supposed to mean by the age of 14 or exactly at the age of 14, if so it should be stated to reflect that. If it is at exactly the age of 14, the need to contextualize and explain why it may be so.

3.     Line 62: Authors describe OCD a “causing a highly disabling condition for the patients and the people around them” The question is if OCD really is a disabling condition for “people around the patients” or are the authors may be referring to burden of care, they should differentiate between this two.

4.     Line 142: I am not sure if emotion suppression is maladaptive per definition. One can think of many situations where that would be an adaptive strategy under given conditions. 

Author Response

Dear reviewer, we very much appreciate all the comments. On this basis we have made changes and marked them in yellow in the manuscript. 

Reviewer 1

This is an interesting study, where the authors try to relate emotion regulation strategies to OCD symptomatology. A general problem I see with the study is that the items listed as emotion regulation strategies (especially the maladaptive ones) might be related to OCD itself, and not entirely independent of it. It may be that participants may be restricted in using a certain emotion regulation strategy by OCD symptom severity. Not that they have preferentially this or that emotion regulation style but they might be restricted in using this or that emotion regulation strategy by the diagnosis itself. In other words, the reasoning might be circular. 

Minor points

  1. Line 9 abstract: The authors write in the abstract “Emotion regulation refers to the management of one's emotions by cognitive strategies.” But not all emotion regulation strategies are cognitive. There is also implicit emotion regulation, suppression, avoidance are other strategies may not always be cognitive strategies. They qualify this later in the text, but it should be reformulated here to reflect the complexity.  

We have changed that phrase: “Cognitive emotion regulation refers to the management of one's emotions through cognitive strategies

  1. Line 54: The authors write on line 54 that “The average age of the onset of OCD is 19 years old, and around 25% of cases frequently begin at the age of 14 years” Is this supposed to mean by the age of 14 or exactly at the age of 14, if so it should be stated to reflect that. If it is at exactly the age of 14, the need to contextualize and explain why it may be so.

We have changed “at the age” for “before the age”.

  1. Line 62: Authors describe OCD a “causing a highly disabling condition for the patients and the people around them” The question is if OCD really is a disabling condition for “people around the patients” or are the authors may be referring to burden of care, they should differentiate between this two.

We have changed the writing. No it says: “Also, there are usually fluctuations, and it is exacerbated in stressful periods, causing a highly disabling condition for the patients and can negatively affect those around them”.

  1. Line 142: I am not sure if emotion suppression is maladaptive per definition. One can think of many situations where that would be an adaptive strategy under given conditions.

Earlier we had commented that "There is some agreement that cognitive reappraisal is a more effective or adaptive strategy than expressive emotional suppression, which can be counterproductive". We have thought that it would be clearer if in the paragraph the reviewer comments on we added "expressive". Now it says: “Likewise, maladaptive emotion regulation, such as that which occurs with expressive emotional suppression, would be related to greater cognitive rigidity and obsessive-compulsive symptomatology.”

Please find attached the document with the point-by-point response to both reviewers. 

Reviewer 2 Report

I have read the article titled "Relationship between Cognitive Strategies for Emotion Regulation and Subtypes of Obsessive-Compulsive Symptomatology in Adolescents" with great interest. It is commendable to see that your research aims to investigate the relationship between cognitive strategies for emotion regulation and subtypes of obsessive-compulsive symptomatology in adolescents. The findings presented in your research were informative considering the scarcity of studies in this age group, and I would like to offer some constructive feedback to enhance the quality of your research further.

1) I noted that the introduction section was lengthy and focused on the epidemiology of OCD. I suggest that you could consider summarizing this section and shifting the focus to the prevention of OCD in healthy adolescents. Why do you talk about subtypes in the title of this paper? No typology is offered; rather your study is based on a dimensional assessment of symptoms. Furthermore, It might be beneficial to include and illustrate some references in the introduction (e.g., from [56] onward), because they are crucial for interpreting the results and discussion.

2) I would like to know more about how the exclusion criteria were verified. It would be helpful to provide more information regarding the verification process, as it would enhance the transparency and reliability of the study.

3) I am aware that the OCI-CV questionnaire employs cut-off scores to identify risk profiles. For instance, scores ≥ 11 points on the OCI-CV denote caseness (see Rough et al., 2020). Could you provide descriptive data on how many adolescents exceeded clinical cut-offs? It would be helpful to consider whether it is necessary to control for this variable in the analysis, given that the sample mean was around 12. (See also if Cervin et al. 2022 suggested cut-offs for subdimensions)

Rough, H. E., Hanna, B. S., Gillett, C. B., Rosenberg, D. R., Gehring, W. J., Arnold, P. D., & Hanna, G. L. (2020). Screening for pediatric obsessive–compulsive disorder using the obsessive–compulsive inventory-child version. Child Psychiatry & Human Development, 51, 888-899.

Cervin, M., Garcia-Delgar, B., Calvo, R., Ortiz, A. E., & Lazaro, L. (2022). Symptom dimension breakpoints for the obsessive-compulsive inventory-child version (OCI-CV). Child Psychiatry & Human Development, 1-8.

4) I would like to inquire whether the assumptions of the multiple regression model have been verified. The model makes several assumptions, including linearity, homoscedasticity, normality, and no multicollinearity. It is crucial to ensure that these assumptions are met to enhance the validity and reliability of the results.

5) Since the study involves two sets of variables and a cross-sectional data collection, I believe that a canonical analysis would be more appropriate and effective than a regression model. As an example, I suggest consulting the work of Adarkwah & Hirsch (2020) or other studies that employ this type of analysis to set up the analyses and report results.

Adarkwah, C. C., & Hirsch, O. (2020). The association of work satisfaction and burnout risk in endoscopy nursing staff—A cross-sectional study using canonical correlation analysis. International Journal of Environmental Research and Public Health, 17(8), 2964.

6) In the discussion section, you mentioned that self-reports do not allow clinical diagnoses to be obtained, which can lead to social desirability bias and false answers. I would like to suggest that you also should consider the potential for common method bias in a cross-sectional study that relies on self-report measures only. It would be helpful to evaluate different methods to assess common method bias, such as Harman's single-factor test or including a marker variable.

In conclusion, I want to reiterate that your study is informative. Addressing these concerns and suggestions will further enhance the quality and clarity of your research. I hope that you find my feedback helpful and useful.

Author Response

Please see the attachment with our point-by-point response to the reviewer's comments

Round 2

Reviewer 1 Report

The authors should address the issue raised about potential circular reasoning, that the maladaptive cognitive regulation strategies they discuss in relation to OCD may either be predisposing factors for the illness or a consequence of the illness. I have not see that this issue is addressed.

Author Response

Please find attached the document with the response to the editor and reviewers.

Reviewer 2 Report

Dear Authors,

I had the chance to review your revision and would like to thank you for addressing the issues raised in my initial review in such a thoughtful and open manner. Your remarks show that you have worked hard to improve the manuscript. However, I must confess that some of the changes have left me desiring more. I think you could still do better with the overall caliber of your study. Your manuscript will be improved and any uncertainties will be cleared up as a result of the following replies to your rebuttal letter. Looking forward to reading your work after revision and observing your advancements.

R1) Thank you for your response. It is critical to have a thorough understanding of the topic, and your approach to discussing OCD in the introduction definitely achieves this. However, it is also critical to strike a balance between giving the necessary context and keeping the paper focused on the primary research question. Given the importance of prevention in healthy adolescents, you could keep the most important information about OCD and its prevalence while condensing some of the epidemiological details. Finally, including more references in the introduction, especially from source [56], would lay a strong foundation for the findings and discussion sections. By doing this, you make sure that readers have access to enough background information to understand the importance of your findings and the ramifications of your study.

R2) Thank you for providing more details about how the exclusion criteria were applied in your research. It is critical that all readers comprehend the process used to select participants and how the data was gathered. You provide a better picture of the participant selection process and ensure that readers are aware of the measures taken to maintain the integrity of the data gathered by describing the role of educational centers and guidance teams in verifying the exclusion criteria. This information should be included in the manuscript to improve the transparency and reliability of your research.

R3) I appreciate your concerns about the large proportion of participants who exceeded the clinical cut-off suggested by Rough et al. (2020). It is therefore critical to address these concerns in your manuscript's limitations section to ensure that readers are aware of the potential consequences of your study's findings and conclusions.

For example, you could discuss one or more of the following points:

There might be cultural and linguistic differences. Rough et al. (2020) derived their cut-off scores from a clinical group in the United States. Given the cultural, linguistic, and demographic differences between the two groups, these cut-offs may not be directly relevant to your sample. It is worth emphasizing the importance of future research to set appropriate cut-off scores for the Spanish population.

There might be issues with cross-cultural instrument validity. The fact that a significant proportion of your sample exceeded the clinical cut-off might raise concerns about the OCI-CV questionnaire's validity in your particular community. You could discuss this limitation and talk about the need for additional validation studies of the OCI-CV in comparable populations to ensure its reliability and validity for determining caseness.

It is also possible that your sample has unique characteristics that contribute to the high percentage of participants exceeding the clinical cut-off. This could be due to sampling methods, regional influences, or other undisclosed motives. It is critical to discuss these potential factors and the implications for the generalizability of your results

If you address these concerns in the limitations section, you will provide a more balanced and transparent discussion of your study's findings and their potential consequences. This will not only help readers understand the context of your results, but it will also inspire future research to address these limitations and further validate the instrument in various populations.

R4) I value your thorough response and the steps you undertook to test the multiple regression model’s assumptions. It’s reassuring to know that the majority of your assumptions were fulfilled, which increases the validity and reliability of your findings. I am satisfied with your response, and I think that including this information in the manuscript will help readers comprehend the statistical analyses performed in your study.

R5) I acknowledge your concerns, and while I respect your viewpoint, I would like to emphasize that the suggested analysis does hold relevance in the context of this study. I  would like to reiterate the reasons why canonical correlation analysis (CCA) might be more appropriate for this study, given its cross-sectional design and the examination of relationships between two sets of variables.

a)        CCA is intended to investigate connections between two groups of variables. (in this case, Set 1 [OCD subscales] and Set 2 [emotional regulation factors]). This is in line with the study's goal of looking into the connection between cognitive strategies for emotion regulation and particular dimensions of obsessive-compulsive symptomatology in adolescents.

b)        CCA identifies linear combinations of variables from Sets 1 and 2 that maximize the correlation between the two sets. This is useful for comprehending the complex relationships between OCD symptomatology dimensions and emotional regulation techniques. Furthermore, the 0.59 correlation of the first canonical variate is lower than the common method factor bias threshold. This observation not only supports the relevance of the suggested analysis but also helps address concerns regarding potential biases in your study.

c)        Unlike regression models, CCA allows you to evaluate the overall relationship between the two constructs (OCD symptomatology and cognitive emotion regulation) without specifying dependent and independent variables. Due to the cross-sectional design of your research, CCA can provide a more complete understanding of the associations between emotional regulation techniques and OCD symptoms without making assumptions about their causal relationships.

While I acknowledge that the authors may have specific reasons for choosing a regression model, I believe that considering canonical correlation analysis could provide valuable insights into the relationships between the two sets of variables.

Here are some thoughts that came to my mind after reading your "pantagruelic" summary of CCA findings.

Obsession, doubt, and neutralization were the primary obsessive-compulsive symptoms reported by "normative" adolescents. Catastrophism, self-blame, and reappraisal were the primary emotional regulation strategies in the smaple studied.

CCA emphasizes the importance of reappraisal as an adaptive emotional regulation strategy that seems to be related to a decrease in obsessive-compulsive symptoms, one can focus on positive coping strategies. Reappraisal, for example, might lead to a change in adolescents’ emotional reactions. This positive coping technique can help adolescents manage and reduce the severity of obsessive-compulsive symptoms.

Catastrophism and self-blame appear to be maladaptive emotional regulation strategies linked with increased obsessive-compulsive symptoms in CCA. These strategies can exacerbate emotional distress and increase the severity of obsessive-compulsive symptoms. Thus, CCA suggests that catastrophism and self-blame might be prioritized in health-promoting interventions aimed at preventing OCD in healthy adolescents.

CCA also suggests other emotional regulation strategies that may have a lesser but still significant effect on the relationship between emotional regulation and obsessive-compulsive symptoms. Despite smaller statistical coefficients, focusing and planning can help alleviate obsessive-compulsive symptoms.

Finally, the findings of this CCA highlight the importance of encouraging adaptive emotion regulation strategies such as reappraisal, focusing, and planning in order to manage and reduce obsessive-compulsive symptoms. Furthermore, addressing and reducing maladaptive strategies such as feelings of catastrophism and self-blame can help adolescents reporting OCD symptoms better their mental health.

P.S: If you feel that canonical correlation analysis (CCA) does not suit your study's needs, you might consider using network analysis to obtain similar information without relying on latent variables. While I do not explicitly recommend this strategy (because it might be too demanding at this point), I would like to stress that traditional regression methods may look outdated and may provide limited clinical insights, particularly in the context of cross-sectional studies. Predictive models, on the other hand, are typically more successful in predicting future outcomes and may be more useful for longitudinal designs.

R6) I understand that you did not perform a common method bias check in your study. However, based on the information provided, it appears that the first canonical factor did not account for the majority of the variance. This finding lends support to the notion that common method factor bias might not be a significant concern in your study. Furthermore, this observation strengthens the argument for using canonical analysis either in addition to or as a replacement for regression analyses in your study. By incorporating canonical analysis, you could more effectively explore the relationships between the two sets of variables and potentially address the issue of common method bias in a more comprehensive manner.

Author Response

(The authors gave the same response as above.)
